# Differentiable Optimization of Generalized Nondecomposable Functions using Linear Programs

## Abstract

We propose a framework which makes it feasible to directly train deep neural networks with respect to popular families of task-specific non-decomposable performance measures such as AUC, multi-class AUC, $F$-measure and others. A common feature of the optimization model that emerges from these tasks is that it involves solving a Linear Programs (LP) during training where representations learned by upstream layers influence the constraints. The constraint matrix is not only large but the constraints are also modified at each iteration. We show how adopting a set of influential ideas proposed by Mangasarian for 1-norm SVMs – which advocates for solving LPs with a generalized Newton method – provides a simple and effective solution. In particular, this strategy needs little unrolling, which makes it more efficient during backward pass. While a number of specialized algorithms have been proposed for the models that we describe here, our module turns out to be applicable without any specific adjustments or relaxations. We describe each use case, study its properties and demonstrate the efficacy of the approach over alternatives which use surrogate lower bounds and often, specialized optimization schemes. Frequently, we achieve superior computational behavior and performance improvements on common datasets used in the literature.

## 1 Introduction

Commonly used losses such as cross-entropy used in deep neural network (DNN) models can be expressed as a sum over the per-sample losses incurred by the current estimate of the model. This allows the direct use of mature optimization routines, and is sufficient for a majority of use cases. But in various applications ranging from ranking/retrieval systems to class imbalanced learning, the most suitable losses for the task do not admit a "decompose over samples" form. Examples include Area under the ROC curve (AUC), multi-class variants of AUC, $F$-score, Precision at a fixed recall (P@R) and others. Optimizing such measures in a scalable manner can pose challenges even in the shallow setting. Since the compromise involved in falling back on a decomposable loss when a non-decomposable objective may be more appropriate for the task at hand can range from negligible to concerning depending on the application, the last few years have seen a number of interesting approaches proposed to efficiently deal with such structured and non-decomposable losses. To this end, recent algorithms for AUC maximization have been developed based on convex surrogate losses Liu et al. (2018); Natole et al. (2018) in a linear model or in conjuction with a deep neural network Liu et al. (2019) as well as stochastic and online variations (Ataman et al. (2006); Cortes & Mohri (2004); Gao et al. (2013); Liu et al. (2018; 2019)) are available. Methods for measures other than the AUC have also been studied – exact algorithms for optimizing $F$-score Nan et al. (2012); Dembczynski et al. (2011), optimizing average precision through direct optimization Song et al. (2016), scalable methods for non-decomposable objectives Eban et al. (2017); Venkatesh et al. (2019) and using a structured hinge-loss upper bound to optimize average precision and NDCG Mohapatra et al. (2018). Recently, the AP-Perf method Fathony & Kolter (2020) showed how custom non-decomposable performance measures can be conveniently incorporated into differentiable pipelines.

It is known that a number of these non-decomposable objectives can be expressed in the form of an integer program that can be relaxed to a linear program (LP). Earlier approaches adapted methods for structured SVMs Joachims et al. (2009) or cutting plane techniques Yue et al. (2007) and were interesting but had difficulty scaling to larger datasets. More recently, strategies have instead focused

on the stochastic setting where we operate on a mini-batch of samples. A common strategy, which is generally efficient, is to study the combinatorial form of one or more losses of interest and derive surrogate lower bounds. These are then tackled via specialized optimization routines. The arguably more direct alternative of optimizing as a module embedded in a DNN architecture remained difficult until recently – but OptNet Amos & Kolter (2017) and CVXPY now offer support for solving certain objectives as differentiable layers within a network. Further, methods based on implicit differentiation have been applied to various problems, showing impressive performance.

Our approach is based on the premise that tackling the LP form of the non-decomposable objective as a module within the DNN, one which permits forward and reverse mode differentiation and can utilize in-built support for specialized GPU hardware in modern libraries such as PyTorch, is desirable. First, as long as a suitable LP formulation for an objective is available, the module may be directly used. Second, based on which scheme is used to solve the LP, one may be able to provide guarantees for the non-decomposable objective based on simple calculations (e.g., number of constraints, primal-dual gap). The current tools, however, do not entirely address all these requirements, as we briefly describe next. A specific characteristic of the LPs that arise from the losses mentioned above is that the constraints are modified at each iteration – as a function of the updates to the representations of the data in the upstream layers. Further, the mini-batch of samples changes at each iteration. Solvers within CVXPY, are effective but due to their general-purpose nature, rely on interior point methods. OptNet is quite efficient but designed for quadratic programs (QP): the theoretical results and its efficiency depends on factorizing a matrix in the quadratic term in the objective (which is zero/non-invertible for LPs). The primal-dual properties and implicit differentiation for QPs do not easily translate to LPs, and efficient ways of dealing with constraints that are iteratively updated are not widely available at this time. In principle, of course, backpropagating through a convex optimization model (and in particular, LPs) is not an unsolved problem. For LPs, we can take derivatives of the optimal value (or the optimal solution) of the model with respect to the LP parameters, and this can be accomplished by calling a powerful external solver. Often, this would involve running the solver on the CPU, which introduces overhead. . The ideas in Meng et al. (2020) are relevant in that the optimization steps for the LP are unrolled and only involve simple linear algebra operations but the formulation is applicable when the number of constraints are about the same as the number of variables – an assumption that does not hold for the models we will study.

In §3, we show that the modified Newton's algorithm in Mangasarian (2004) can be used for deep neural network (DNN) training in an end-to-end manner without requiring an external solvers where support for GPUs remains limited. Specifically, by exploiting self-concordance of the objective, we show that the algorithm can converge globally *without* line search strategies. On the practical side, we analyze the gradient properties, and some modifications to improve stability during backpropagation. We show that this scheme based on Mangasarian's parametric exterior penalty formulation of the primal LP can be a computationally effective and scalable strategy to solve LPs with a large number of constraints.

## 2 NONDECOMPOSABLE FUNCTIONS AND CORRESPONDING LP MODELS

We first present a standard LP form and then reparameterize several generalized nondecomposable objectives in this way, summarized in Table 5 in the appendix. We start with the binary AUC, extend it to multi-class AUC, and then later, show a ratio objective, $F$-score. Some other objectives are described in the appendix.

### 2.1 NOTATIONS AND GENERALIZED LP FORMULATION

**Notations.** We use the following notations:
**(i)** $n$: number of samples used in training.
**(ii)** $X \in \mathbb{R}^{n \times d}$ : the explanatory features fed to a classifier (e.g., parameterized by $\mathbf{w}$);
**(iii)** $f(x_i)$ (or $f(i)$): a score function for the classifier where $x_i \in X$ such that $f(X) = \mathbf{w}X$;
**(iv)** $Y \in \{0, 1\}$: target label and $\hat{Y} \in \{0, 1\}$: predicted label for binary classification, both in $\mathbb{R}^n$;
**(v)** $\phi(\cdot)$: non linear function applied on $f(X)$;
**(vi)** $A \otimes B$: Kronecker product of matrices $A$ and $B$.
**(vii)** $I_r$: Identity matrix of size $r$ and $\mathbb{1}$ is the indicator function.
**(viii)** $\mathbf{B}_{k,-}$ (and $\mathbf{B}_{|,k'}$) gives the $k$-th row (and $k'$-th) column of $\mathbf{B}$.

**LP formulation.** We consider a general linear program (LP) that contains nonnegative variables as well as inequality and equality constraints. The form of the LP is given as

$$\max_{u,v} g^T u + h^T v \quad \text{s.t} \quad Eu + Fv \leq p, \quad Bu + Gv = q \quad u, v \geq 0 \tag{1}$$

We can write it more succinctly as

variable $x = [u\ v]$; coefficient $c = [-g\ -h]$; constraints $A = \begin{bmatrix} E & B & -B \\ F & G & -G \end{bmatrix}^T$; constants $b = [p\ q\ -q]^T$

The corresponding primal LP becomes $\min_x c^T x$ s.t $Ax \leq b,\ x \geq 0$.

## 2.2 MAXIMIZING AUC

The Area under the ROC Curve (AUC) calculates the probability that a classifier $f(\cdot)$ will rank a randomly chosen positive sample higher than a randomly chosen negative sample. Since AUC varies between 0 and 1, where 1 represents all positives being ranked above the negatives. AUC may be estimated using the Wilcoxon-Mann-Whitney (WMW) Statistic Hanley & McNeil (1982), as

**Definition 2.1** (AUC). Let $n$ be the number of samples. Let $X_+$ (and $X_-$ resp.) be the set of positive (and negative resp.) samples such that $|X_+| + |X_-| = n$ where $|\cdot|$ is the cardinality of the set. Then, AUC is given as $\left( \sum_{i=1}^{|X_+|} \sum_{i=1}^{|X_-|} \mathbb{1}_{f(x_i) > f(x_j)} \right) / \left( |X_+|\,|X_-| \right)$ for $x_i : i \in \{1, \cdots, n\}$.

Here, we follow Ataman et al. (2006) to calculate the AUC based on the WMW statistic as follows.

$$\min_{z_{ij}} \sum_{i=1}^{|X_+|} \sum_{j=1}^{|X_-|} z_{ij} \quad \text{s.t.} \quad f(x_i) - f(x_j) \geq \epsilon - z_{ij} \quad \text{where } x_i \in X_+, x_j \in X_-; \quad z_{ij} \geq 0, \tag{2}$$

where $\epsilon$ is a given constant. The model in (2), maximizes AUC indirectly, by minimizing the number of pairs (one each from the positive and negative classes) where the positive sample is not ranked higher than the negative sample: so, the number of zero entries in $z$ equals the number of pairs where this condition is not true. To compute the AUC, we do:

$$\text{AUC} = \left( n - \|z\|_0 \right) / \left( |X_+|\,|X_-| \right) = \left( n - \sum_i \sum_j \epsilon^{-1} \text{relu}(0, -z_{ij} + \epsilon) \right) / \left( |X_+|\,|X_-| \right). \tag{3}$$

If $z_{ij}$ is 0, then $\epsilon^{-1}\text{relu}(0, -z_{ij} + \epsilon)$ equals 1. Otherwise $z_{ij} > 0$, it follows from the first constraint in (2), that $z_{ij} \geq \epsilon$, so $\epsilon^{-1}\text{relu}(0, -z_{ij} + \epsilon)$ equals 0. Observe that in (2), the number of constraints is $|X||X_-|$, which is quadratic in $n$.

## 2.3 MAXIMIZING MULTI-CLASS AUC

An extension of AUC to the multi-class setting, $\text{AUC}_\mu$, is defined in Kleiman & Page (2019). The $\text{AUC}_\mu$ objective optimizes an indicator matrix computed on the orientation function, $O_{i,j}$ defined as,

**Definition 2.2** (Orientation Function; Kleiman & Page (2019)). Assume we have $K$ classes $\{y_1, \cdots, y_K\}$. Let $\mathbf{f}(x_i, -) \in \mathbb{R}^K$ indicate the model's prediction on $x_i$ for each of the $K$ classes (class-specific probability)[1]. Let $x_i^*$ provide the index of $x_i$'s true class label. Let $\mathbf{P} \in \mathbb{R}^{k \times k}$ be a partition matrix where $\mathbf{P}_{k,k'}$ is the cost of classifying a sample as class $k$ when it should be $k'$. Define $\mathbf{v}_{kk'} = \mathbf{P}_{k,-} - \mathbf{P}_{k',-}$ and $\widetilde{\mathbf{v}} = \mathbf{v}_{x_i^* x_j^*} \in \mathbb{R}^K$. Then, $O_{i,j} = (\widetilde{\mathbf{v}}_{x_i^*} - \widetilde{\mathbf{v}}_{x_j^*})(\langle \widetilde{\mathbf{v}}, \mathbf{f}(x_i, -) \rangle - \langle \widetilde{\mathbf{v}}, \mathbf{f}(x_j, -) \rangle)$.

Here, $\mathbf{v}_{kk'}$ ranks the instances by their cost difference between assignments to class $k$ and $k'$. For two classes, say indexed by 1 and 2, $\mathbf{v}_{1,2} \cdot \mathbf{f} = 0$ is the decision boundary between the classes. The ranking is correct for $x_i$ and $x_j$ (with correct labels 1 and 2 resp.) if the orientation of the points w.r.t. the decision boundary is the same as those of the class labels (converted to one-hot vectors).

Then $\text{AUC}_\mu = \sum_{k < k'} g_{k,k'} \sum_{i \in D_k, j \in D_k'} \mathbb{1}_{O_{i,j} \geq 0}$, where $D_k$ is the index set of samples having same class label $k$ and $g$ is a constant derived from class sizes. In formulating the linear program, we observe that the model is dictated on how $\mathbf{P}$ is defined. One way is to set $\mathbf{P}(k, k) = 0\ \forall k$ and 1 for all other entries. We can also define a $\mathbf{P}$ with arbitary entries or formulate AUC in a one-vs-all setting. Here, we show the model in the first case (the other two cases are discussed in

---

[1]extension of $f(.)$ to multi-class setting

Appendix A.1). Similar to our previous formulation, the goal is to minimize the sum of negative values of the orientation function, rather than maximize the sum of positive values. Let $\widetilde{\mathbf{f}}(i,j) = \mathbf{f}(x_i, x_i^*) - \mathbf{f}(x_j, x_i^*) + \mathbf{f}(x_j, x_j^*) - \mathbf{f}(x_i, x_i^*)$. Then the LP formulation is.

$$\text{AUC}_\mu^{\text{bin}} : \min_{z_{ij}} \sum_{i=1}^n \sum_{j=1:x_i^*<x_j^*}^n z_{ij} \quad \text{s.t.} \quad \widetilde{\mathbf{d}}_{ij}\widetilde{\mathbf{f}}(i,j) \geq \epsilon - z_{ij}, \quad \forall i,j : x_i^* < x_j^* \qquad z_{ij} \geq 0 \quad (4)$$

Here $\widetilde{\mathbf{d}}_{ij} = \widetilde{\mathbf{v}}_{x_i^*} - \widetilde{\mathbf{v}}_{x_j^*}$. This can be seen as an extension of our binary AUC model, where the $z_{ij}$ is the ranking between a pair of points defined for multiple classes.

## 2.4 MAXIMIZING $F$-SCORE

The $F$-score (or $F$-measure) is a representative of objectives expressed as ratios of some combination of True positives (TP), False positives (FP), True negatives (TN) and False negatives (FN). The general form of the ratio functions and formulations for other objectives is in the Appendix A.2. Specifically, $F$-score is defined as follows:

**Definition 2.3** ($F$-score). $F\text{-score} = \frac{2(\text{Precision}\times\text{Recall})}{\text{Precision}+\text{Recall}} = \frac{2\text{TP}}{2\text{TP}+\text{FP}+\text{FN}} = \frac{2(Y^T \times \hat{Y})}{\mathbf{1}^T Y + \mathbf{1}^T \hat{Y}}$

The second part of the equality comes from simplying the precision ($\frac{TP}{TP+FP}$) and recall ($\frac{TP}{TP+FN}$) based formula Dembczynski et al. (2011). The last part is obtained by replacing TP with ($Y^T \times \hat{Y}$), FP with $(1-Y)^T \times \hat{Y}$ and FN with $(Y)^T \times (1-\hat{Y})$ as functions of $Y$ and $\hat{Y}$. This leads to the following integer fractional optimization model,

$$F\text{-score} = \max_{\hat{Y}} \frac{\mathbf{c}^T \hat{Y}}{1^T\hat{Y}+b} \quad \text{s.t.} \quad \hat{Y}_i \in [0,1], \ i=1,\ldots,n \text{ where } c = 2Y \text{ and } b = \sum_{i=1}^n Y_i. \quad (5)$$

To solve this, we first relax the constraint on $\hat{Y}$ and reformulate the model as the following LP, by introducing two variables $z \in R^n$ and $t \in R^1$ where $z = \frac{b\hat{Y}}{1^T\hat{Y}+b}, t = \frac{b}{1^T\hat{Y}+b}$ and $i \in \{1,\cdots,n\}$:

$$\max_{z,t} \frac{\mathbf{c}^T z}{b} \quad \text{s.t} \quad \underbrace{1^T z + bt = b}_{(a)}; \ \underbrace{z_i \leq t}_{(b)}; \ \underbrace{\phi(f(x_i))t \leq z_i \leq (1+\phi(f(x_i)))t}_{(c)}; \ 1 \geq z_i, t \geq 0,$$

**Remark 1.** *(a) ensures the appropriate relation between $z$, $t$ and $\hat{Y}$ and is essentially a reformulaton of the ratio objective as a linear function with a fixed denominator. $\hat{Y}$ is recovered from the solution to the linear program by computing $\hat{Y}_i = \frac{z_i}{t}$ when $t > 0$ and $\hat{Y}_i = z_i$ otherwise. (b) sets an upper bound for $\hat{Y}_i \leq 1$. (c) ties the output of the previous layer $\phi(f(x_i))$ (a classifier score for $\hat{Y}_i$, see the definition in Section 2.1) as a input in this layer. Assume $\phi(.) \in \{-1,1\}$ (ensured if $\phi$ is sigmoid or tanh) is the indicator of the class label (based on sign). We want $\hat{Y}_i \geq \phi(f(x_i))$ and $\hat{Y}_i \leq 1 + \phi(f(x_i))$. If $\hat{Y}_i$ is $\{0,1\}$, these two constraints ensures that $\hat{Y}_i = 0$ when $\phi(f(x_i)) \leq 0$ and $\hat{Y}_i = 1$ when $\phi(f(x_i)) > 0$. When $\hat{Y}_i$ is relaxed and replaced with $z$ and $t$, we get the equivalent form (c).*

This model imposes $4n$ constraints for $n$ samples. Since this is a maximization, a solution to the LP, $O_{\hat{Y}}$, is an upper bound on the integer objective $opt^*$, and serves as the loss.

## 3 BACKPROPAGATION VIA FAST EXTERIOR PENALTY OPTIMIZATION

Unlike traditional feedforward networks, where the output of each layer is a relatively simple (though non-linear) function of the previous layer, a LP layer must solve a constrained optimization problem, therefore implementing scalable and efficient backpropagation schemes that minimizes computational overhead requires more care and as reviewed in §1, is an active topic of research. This problem is, of course, not unique to LPs and manifests in differentiable sorting Mena et al. (2018) and formulating quadratic or cone programs Amos et al. (2017). One may unroll gradient descent steps Amos et al. (2017); Goodfellow et al. (2013); Metz et al. (2016) or use projections Zeng et al. (2019). Recently

Agrawal et al. (2019) introduced a package for differentiable constrained convex programming, which includes LPs as a special case. For LPs, Meng et al. (2020) presents an unrolled scheme and Blondel et al. (2020) shows that we can differentiate through LP formulations of sorting/ranking *exactly* by using smooth approximations of projection steps. Berthet et al. (2020) describes an interesting approach where one computes approximate gradients through ranking/shortest path problems by stochastic perturbation techniques.

**Remark 2.** *Some previous works Zeng et al. (2019) have considered LPs where the constraints are deterministic (for a fixed input dimension), i.e., do not depend on the data $X$, which is different from the LPs in §2.2–2.4.*

Note that such perturbation techniques in Berthet et al. (2020) are applicable to our LPs as well. For example, the Fenchel Young losses as defined in Berthet et al. (2020) is attractive because there is no need to compute the Jacobian. From the implementation standpoint, one could simply think of the backward pass as a function given the input and output of the forward pass. But the gradient expressions of the losses is an expectation and hence may require multiple calls to a LP solver in order to approximate the expectation. although parallelization and warm starts were shown to alleviate this dependency by sampling in parallel.

**Rationale.** Consider a LP with a large $m$ number of constraints in fixed dimensions $n$ ($n \ll m$). This assumption holds in all formulations in §2. This is because we assume that the architecture is fixed whereas minibatch size depends on the complexity of the task (stable gradient or when noise in gradient is high). Hence, solving such LPs using off-the-shelf solvers as in Berthet et al. (2020) may slow down the training process. The strategy in Agrawal et al. (2019) does offer benefits over Amos & Kolter (2017) for sparse QPs. Our strategy is to run Mangasarian's Newton's method on an exterior penalty function of the LP. There are two advantages: **(i)** during **forward** pass, quadratic local convergence of Newton's method indicates that unrolling the method may be a reasonable choice; and further **(ii)** based on the relationship between dual and primal variables, and the exactness of the exterior penalty we can show that **backward** pass is independent of $m$. We will discuss both these results and some modifications to deal with discontinuous Hessian (and its inverse) that is required for Newton's method. A similar approach is proposed in Amos & Kolter (2017) in which Primal-Dual Interior Point methods with implicit differentiation is used for differentiation purposes. But the exterior penalty in (6) satisfies an interesting property: primal and dual solutions are related by a *closed form* expression which can be exploited for efficient backpropagation.

### 3.1 FORWARD PASS USING NEWTON'S ALGORITHM

A key requirement for fast automatic (forward or reverse mode) differentiation is that we can perform the *forward* pass efficiently. In our setting, we seek to solve and backpropagate through an LP. We will focus on reverse mode differentiation since it is the most suitable for DNN training.

Given a primal LP, for a fixed accuracy $\epsilon > 0$, Mangasarian (2004) solves an unconstrained problem,

$$\min_y g(y) := \frac{1}{2} \|\sigma(Ay - b)\|^2 + \epsilon c^T y, \tag{6}$$

where $\sigma(\cdot) = \max(\cdot, 0)$ represents the elementwise relu function. A modified Newton's method can be use to solve equation 6 that performs the following iterations:

$$y = y + \lambda d \text{ where } d = \tilde{H}(y)^{-1} \nabla g(y) := (\nabla^2 g(y) + \rho I)^{-1} \nabla g(y). \tag{7}$$

In large scale settings of $A, b$, such Newton methods are known to perform empirically better than gradient descent Mangasarian (2006); Keerthi et al. (2007). We will discuss if this holds for our purposes shortly.

**Why is Newton's method applicable for minibatches?** In general, the convergence of Newton's method depends strongly on initialization, i.e., we can only provide local convergence results. However, this is not the case for our problems since in our examples, either the level sets are bounded from below; or the feasible set is compact, as noted in Mangasarian (2004). There are two reasons why the result, by itself, is insufficient for our purposes: **(i)** it assumes that we can perform line search to satisfy Armijo condition; **(ii)** even with line search, the result does not provide a *rate* of convergence. In DNN training, such line search based convergence results can be prohibitively expensive. The main difficulty is handling the discontinuity in the Hessian. As a remedy, we use self concordance of

(6) to guarantee global convergence of (7) iterations for the exterior penalty formulation in (6). To do so, we first show a result (proof in Appendix) that characterizes the discrepancy between the actual Hessian of (6) and the modified one in (7) when $A$ is randomly distributed.

**Lemma 1.** *Assume that $A$ is a random matrix, and fix some $y \in \mathbb{R}^n$. Then with probability one, g in equation 6 is constant (given by $\tilde{H}, \nabla g(y)$) over a sufficiently small neighborhood of y.*

Intuitively, Lemma 1 states that with probability one, each $y$ has a neighborhood in which the Hessian is constant. In addition, the modified Hessian is nonsingular at all points (in particular the optimal $y^*$), and so we can then show the following global convergence result.

**Theorem 2.** *Newton's method converges globally at a linear rate with local quadratic convergence.*

*Proof.* (Sketch) First, we use the fact that the objective in equation 6 is piecewise quadratic, and hence self concordant. Second, observe that the possible choice of Hessian are finite, and so we can choose $\rho > 0$ so that there is a descent direction *without* line search, that is, there exist a step size $\lambda > 0$ such that $\lambda \nabla g(x)^T d < 0$. Finally, we use Theorem 4.1.12 in Nesterov (2013) to claim the result. $\square$

Please see Appendix for the proof. The loss function is almost surely quadratic in the neighborhood of any $y$, thus intuitively suggests local quadratic convergence independent of the starting point.

**Remark 3.** *Convergence in Thm. 2 is guaranteed under standard constraint qualification assumptions. Linear Independent Constraint Qualification (LICQ) is satisfied for AUC, and Multi-class AUC formulations in §2. But the F-score formulation does not satisfy LICQ, hence we need safeguarding principles in the initial iterations (until iterates get close to the optimal solution).*

## 3.2 BACKWARD PASS USING OPTIMAL DUAL VARIABLES AIDED BY UNROLLING

The advantage of optimizing the exterior penalty in (6) is that given an iterate $y_t$, accuracy $\epsilon$, we can extract the optimal dual solution $v_t$ by simple thresholding, that is, $v_t = 1/\epsilon(A^T \sigma(Ay - b))$. By complementarity slackness, the nonzero coordinates of $v_t$ specify the set of active constraints in $Ax \leq b$. So, given an approximate solution $y_t$ such that $\nabla g(y_t) \tilde{H}(y_t)^{-1} \nabla g(y_t) \leq \epsilon$, to compute the primal solution $x^*$, we solve the "active" linear system given by $\tilde{A}\tilde{b}$, where $\tilde{A}$ denotes the active rows of $A$ and the corresponding subvector $\tilde{b}$. Hence, backpropagation through the layer reduces to computing derivatives of $\tilde{A}^{-1}\tilde{b}$ which is simple via automatic differentiation.

**How to choose $\epsilon$?** If we can successfully retrieve the active constraints at the optimal solution, we do not need to store the intermediate iterates $y_t$ at all during the forward pass (memory efficient). However, setting $\epsilon$ correctly can be tricky for arbitrary polyhedra since it depends on the geometric properties such as facets and vertices that may be difficult to enumerate. One possible way to get around this is to use a "burn-in" period in which we increase $\epsilon$ slightly in each iteration (of deep network training) and backpropagate through the unrolled Newton's iterations during this period. Once we see that the convergence profile has stabilized, we can fix $\epsilon$ at that value and start using the complementarity conditions and derive the active linear system $\tilde{A}^{-1}\tilde{b}$ as discussed above.

**How to backpropagate through unrolled iterations?** We assume that the chain rule is applicable up to this LP layer and is $\frac{\partial L}{\partial x}\frac{\partial x}{\partial A}$ (for one of the parameters $A$), and note that it is possible to find $\frac{\partial L}{\partial x}$ (either directly or using a chain rule). Therefore we focus our attention on $\frac{\partial x}{\partial A}$, which involves the LP layer. Indeed, unrolling each iteration in (7) is equivalent to a "sublayer". So in order to backpropagate we have to show the partial derivatives of each operation or step wrt to the LP parameters $A, b$, and $c$.

Our goal is to $\frac{\partial d}{\partial A}$ where $d = Q^{-1}q$, $Q = \tilde{H}$ and $u = \nabla g(y)$. We can use the product rule to arrive at: $\partial d = -\left(u^T Q^{-1} \otimes Q^{-1}\right)\partial Q + Q^{-1}\partial u$. To see this, note that we have used the chain rule to differentiate through the inverse in the first term. The second term is easy to compute similar to the computation of Hessian. For each of these terms we eventually have to compute $\frac{\partial Q}{\partial z}$ or $\frac{\partial u}{\partial z}$ where $z \in \{c, A, b\}$ which can also be done by another application of chain rule. Please see Appendix A.4 for empirical verification of unrolled gradient and the one provided by $\tilde{A}^{-1}\tilde{b}$.

Before proceeding, we should note an issue that comes up when differentiating each step of the unrolled algorithm due to the fact that the Hessian is piecewise linear (constant) as a function of

the input to that particular layer. Here, some possible numerical approximations are needed, as we describe below.

**Remark 4.** *Note that the diagonal matrix term in $\frac{\partial Q}{\partial A}$ is nondifferentiable due to the presence of the step function. However, the step function is a piecewise constant function, and hence has zero derivative almost surely, that is, in any bounded set $S$, $x \in S$, if a ball (of radius $r > 0$,) $B_r(x) \subseteq S$, then the Lebesgue measure of the set of nondifferentiable points on $S$ is zero. Please see Appendix B.3 for a formal justification where we show this by approximating the step function using a sequence of logistic functions with increasing slope parameter at the origin.*

Therefore, in this setting, Remark 4 provides a way to compute an approximate sub-gradient when using Newton's method based LP layers. The function is a piece-wise quadratic function and differentiable everywhere, and the inverse of the Hessian acts as a preconditioner.

*Summary.* Our forward pass involved three steps: 1. finite steps of Newton's method using which we 2. computed the dual variable by a thresholding operation, and 3. finally, to get the primal solution, these dual variables are first used to identify the active constraints followed by solving a linear system. In order to backpropagate through these three steps, we must differentiate through each layer of our procedure including $\tilde{A}^{-1}\tilde{b}$, independent of whether we use unrolling or Danskin's theorem. Using Danskin's theorem in this setting would involve differentiating through the fixed point of the Newton's iterations similar to (regularized) gradient descent iterations considered in the iMAML work Rajeswaran et al. (2019).

## 4 EXPERIMENTS

In this section, we conduct experiments on commonly used benchmarks to show that our framework can be used to optimize multiple different objectives within deep neural networks and lead to performance gain. We start with binary AUC optimization, and then extend to multi class AUC optimization and $F$-score optimization. We also show that nonnegative matrix factorization can be optimized in linear programming form in our framework.

**Optimizing Binary AUC**

We follow the current state-of-the-art work on AUC optimization Liu et al. (2019) to conduct experiments on optimizing AUC score directly with deep neural networks. The baseline algorithms we compare with for binary AUC are cross-entropy loss and two algorithm (PPD-SG and PPD-AdaGrad) from Liu et al. (2019).

**Datasets:** Cat&Dog, CIFAR10, CIFAR100, and STL10. Cat&Dog is a dataset from Kaggle which contains 25000 images of cats and dogs. 80% of the dataset is used as training set and the rest 20% as test set. STL10 is inspired by the CIFAR-10 dataset but with some modifications. Each class in STL10 has fewer labeled training examples than in CIFAR-10. We follow Liu et al. (2019) to use 19k/1k, 45k/5k, 45k/5k, 4k/1k training/validation split on Cat&Dog, CIFAR10, CIFAR100, STL10 respectively.

**Construction of imbalanced datasets:** We construct imbalanced binary classification task by using half classes as positive class and another half as negative class, and dropping samples from negative class by a certain ratio, which is reflected by the positive ratio (the ratio of the majority class to the minority class) in Table 1.

Table 1: Binary AUC optimization results on four benchmark datasets.

| AUC(%) | Cat&Dog | | | | CIFAR10 | | | |
|---|---|---|---|---|---|---|---|---|
| **Positive Ratio** | 91% | 83% | 71% | 50% | 91% | 83% | 71% | 50% |
| Cross-Entropy | 67.6 | 74.6 | 85.1 | 87.4 | 65.2 | 73.3 | 78.1 | **83.7** |
| PPD-SG | **79.1** | **81.5** | 85.5 | 87.1 | 69.8 | 73.9 | **79.1** | 82.6 |
| PPD-AdaGrad | 77.3 | 80.6 | 83.7 | 86.3 | 69.7 | 74.1 | 78.4 | 83.1 |
| Ours | 78.6 | 81.3 | **85.6** | **87.8** | **72.5** | **74.4** | 78.3 | 82.7 |
| AUC(%) | CIFAR100 | | | | STL10 | | | |
| **Positive Ratio** | 91% | 83% | 71% | 50% | 91% | 83% | 71% | 50% |
| Cross-Entropy | 57.8 | 58.4 | 62.2 | 66.3 | 63.5 | 67.1 | 72.7 | 80.8 |
| PPD-SG | 56.5 | 58.9 | 61.6 | 65.2 | **70.7** | 71.6 | 75.1 | 77.4 |
| PPD-AdaGrad | 56.2 | 59.0 | 62.6 | 67.6 | 68.5 | **72.4** | 76.7 | 78.5 |
| Ours | **58.2** | **60.5** | **64.5** | **69.0** | 68.4 | 71.1 | **76.7** | **81.6** |

**Experimental setting.** We use a Resnet-18 He et al. (2016) as the deep neural network for all algorithms. During optimization, the batch size is set to 64. The initial learning rate is tuned in $\{0.1, 0.01, 0.001\}$ and decays $2/3$ at $2k, 10k, 25k$-th iteration. We train $40k$ iterations in total. The $\epsilon$ in

Newton's method is 0.001. We use the same random seed, learning rate and total number of iterations in all of our experiments including multi class AUC and $F$-score experiments.

**Results.** The results are shown in Table 1. We can see that our method slightly outperforms Liu et al. (2019) and outperforms cross-entropy loss by a large margin, especially on imbalanced datasets, where AUC objective shows superiority over cross-entropy loss.

Table 2: Ablation study of $\epsilon$ on Cat&Dog dataset.

| Positive Ratio | 91% | 83% | 71% | 50% |
|---|---|---|---|---|
| Ours($\epsilon = 0.1$) | 71.3 | 77.0 | 84.4 | 87.3 |
| Ours($\epsilon = 0.01$) | 78.6 | 81.3 | 85.6 | 87.8 |
| Ours($\epsilon = 0.001$) | 65.9 | 71.3 | 71.8 | 76.1 |

**Influence of $\epsilon$.** We report the influence of $\epsilon$ on our algorithm in Table 2, where we choose Cat&Dog as an example to test different $\epsilon$. We can see that $\epsilon = 0.1$ gets slightly worse performance than $\epsilon = 0.01$, while $\epsilon = 0.001$ performs much worse. To choose $\epsilon$, we follow the approach proposed by Mangasarian (2004). If for two successive values of $\epsilon_1 > \epsilon_2$, the value of the $\epsilon$ perturbed quadratic function is the same, then it is the least 2-norm solution of the dual. Therefore, we simply choose an $\epsilon$ that satisfies this property, which is chosen to be 0.01 in our experiments.

**Optimizing Multiclass AUC**

We further demonstrate our method for optimizing multiclass AUC. Similar to previous section on binary AUC, we construct imbalanced multiclass datasets by dividing datasets into 3 classes and drop samples from 2 of them and report the one-versus-all AUC (de-noted as AUC$^{\text{ova}}$) and AUC$^{\text{bin}}_\mu$ score . For STL10, we group class $0 - 2, 3 - 5, 6 - 9$ into the three classes, and drop samples from the first two classes. For CIFR100, we group class $0 - 32, 33 - 65, 66 - 99$ into three classes, and also drop samples from the first two classes.

Table 3: Multiclass AUC optimization results on STL10 and CIFAR100. Drop rate is the proportion used when dropping samples from two of three classes.

| AUC$^{\text{ova}}$(%) | CIFAR100 | | | | STL10 | | | |
|---|---|---|---|---|---|---|---|---|
| **Drop rate** | 90% | 80% | 60% | 0% | 90% | 80% | 60% | 0% |
| Cross-Entropy | 54.3 | 59.4 | 62.7 | 63.5 | 66.9 | 68.0 | 74.8 | 81.0 |
| Ours | 58.4 | 59.2 | 64.1 | 65.7 | 72.9 | 72.5 | 75.7 | 82.7 |
| AUC$^{\text{bin}}_\mu$(%) | CIFAR100 | | | | STL10 | | | |
| **Drop rate** | 90% | 80% | 60% | 0% | 90% | 80% | 60% | 0% |
| Cross-Entropy | 55.1 | 60.6 | 65.0 | 64.0 | 68.9 | 69.6 | 75.8 | 82.2 |
| Ours | 60.1 | 61.2 | 66.0 | 67.2 | 76.1 | 74.4 | 77.7 | 84.5 |

**Results.** Results are in Table 3. In addition to one-versus-all AUC metric, we also report the performance in terms of AUC$_\mu$ Kleiman & Page (2019) which is specifically designed for measuring multiclass AUC and keeps nice properties of binary AUC such as being insensitive to class skew. We can see that our method outperforms cross-entropy loss on all four datasets and under all different skewed ratios. Specifically, the performance gain tends to be larger when the dataset becomes more imbalanced.

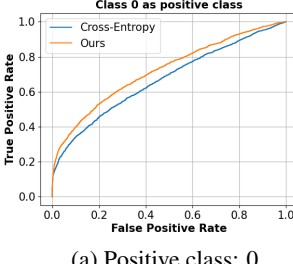
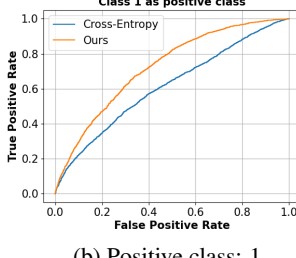
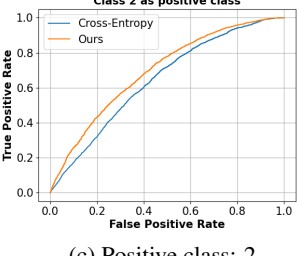

(a) Positive class: 0     (b) Positive class: 1     (c) Positive class: 2

Figure 1: ROC curve of multiclass AUC optimization on STL10 with 90% drop rate. We divide STL10 into 3 classes and use one as positive class and other two as negative class to plot the ROC.

**Optimizing $F$-score**

We show that by directly optimizing $F$-score, we can achieve a better performance on this than when using cross entropy loss. In addition to cross entropy loss, we perform evalua-

Table 4: $F$-score on four datasets.

| $F$-score(%) | Cat&Dog | CIFAR10 | CIFAR100 | STL10 |
|---|---|---|---|---|
| Cross-Entropy | 76.0 | 70.3 | 60.4 | 71.8 |
| CVXPY-SCS | 70.1 | 66.6 | 66.7 | 66.6 |
| AP-Perf | 65.3 | 66.7 | 66.4 | 67.2 |
| Ours | 77.8 | 72.6 | 63.4 | 72.7 |

tions with two other methods that can also directly optimize the $F1$-score. First, we replace our solver with CVXPY-SCS Agrawal et al. (2019), which is a differentiable general purpose linear programming solver; second, we perform comparisons with AP-Perf Fathony & Kolter (2020) which offers differentiable optimization of $F$-score using an adversarial prediction framework. The datasets and our setup to group them into two classes remain the same as in binary AUC section. The results in Table 4 shows that our method generally yields improvement over cross-entropy loss in terms of $F$-score. Note that different from optimizing cross entropy loss, when we optimize $F$-score directly, there exists a local optimal point where assigning all examples to the positive class leads to $F$-score of 66.7%. We see this behavior on CIFAR10, CIFAR100, and STL10.

## 5 CONCLUSIONS

We demonstrated that various non-decomposable objectives can be optimized within deep neural networks in a differentiable way under the same general framework of LPs using a modified Newton's algorithm proposed by Mangasarian. A number of recent papers have studied the general problem of backpropagating through convex optimization modules, and this literature provides several effective approaches although scalability remains a topic of active research. Our work complements these results and shows that the operations needed can be implemented to utilize the capabilities of modern deep learning libraries. While our experimental results suggest that promising results on binary AUC, multi-class AUC and $F$-score optimization within DNNs is achievable, we believe that the module may have other applications where the number of constraints are large and data-dependent.

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

## A APPENDIX

### A.1 LP FORMULATION FOR MULTI-CLASS AUC

One way to extend binary AUC to multi-class is by considering multiple one-versus-all pairs. This leads us to the following formulation:

$$\text{AUC} : \min_{z_{ij}} \sum_{i=1}^{n} \sum_{j=1:x_i^* \neq x_j^*}^{n} z_{ij} \tag{8}$$

$$\text{s.t.} \quad (\mathbf{f}(x_i, x_i^*) - \mathbf{f}(x_j, x_i^*)) \geq \epsilon - z_{ij} \qquad \forall i, j : x_i^* \neq x_j^*, \quad z_{ij} \geq 0$$

In our multi-class AUC experiment, we use this one-versus-all AUC as training loss and report performance in both one-versus-all AUC and $\text{AUC}_\mu^{\text{bin}}$. In addition, we can also consider the setting of $\text{AUC}_\mu$ where $\mathbf{P}$ is set arbitrarily. In this case, the exact terms in orientation function $O$ proposed by Kleiman & Page (2019) can be written as follows:

$$\text{AUC}_\mu^{\text{arbit}} : \min_{z_{ij}} \sum_{i=1}^{n} \sum_{j=1:x_i^* < x_j^*}^{n} z_{ij} \tag{9}$$

$$\text{s.t.} \quad \widetilde{\mathbf{d}}_{ij} \sum_{k=1}^{K} \widetilde{\mathbf{v}}(k)(\mathbf{f}(x_i, k) - \mathbf{f}(x_j, k)) \geq \epsilon - z_{ij} \qquad \forall i, j : x_i^* < x_j^*, \quad z_{ij} \geq 0$$

Note that $\text{AUC}_\mu^{\text{arbit}}$ has the same number of constraints and variables as $\mathbf{AUC}_\mu^{\text{bin}}$. Once the LPs are solved, the loss function is calculated the same way as binary AUC.

### A.2 FORMULATING RATIO OBJECTIVES

In this section, we study a subset of non-decomposable metrics, which are typically expressed as ratios of some combination of True Positive(TP), False Positives(FP), True Negatives(TN) and False Negatives(FN). These can be expressed in a general form as $\frac{a_{11}TP + a_{12}}{a_{21}TP + a_{22}FP + a_{23}FN + a_{24}}$, where $a_{pq}$ are constants/cofficients which if set to 0, means the term is absent and not equal to zero in other cases. This formulation can used to define Fscore, $F_\beta$, Jaccard, IOU and Precision at fixed recall. In the following section, we describe the formulation of Fscore as a representative of this approach, other metrics can be formulated similarly.

Given $Y$ the groud truth, our goal is to compute $\hat{Y}$ both of length $n$, which aligns with $Y$ based on the specific metric. We first show how to write TP, FP, TN and FN wrt to these vectors.

$$TP = Y^T \times \hat{Y} \qquad FP = (1 - Y)^T \times \hat{Y}$$
$$TN = (1 - Y)^T \times (1 - \hat{Y}) \quad FN = (Y)^T \times (1 - \hat{Y})$$

#### A.2.1 FORMULATING $F$-SCORE

The $F$-score or $F$-measure is routinely used as a performance metric for different types of prediction problems, including binary classification and, multi-label classification. Compared to measures like error rate in binary classification and Hamming loss, it enforces a better balance between performance on the minority and the majority classes, and, therefore, it is more suitable in the case of imbalanced data Dembczynski et al. (2011). $F$-score is defined as follows:

$$F\text{-}score(Y, \hat{Y}) = \frac{2P(Y, \hat{Y}) \times R(Y, \hat{Y})}{P(Y, \hat{Y}) + R(Y, \hat{Y})} \tag{10}$$

where $P$ is the measure of precision defined as

$$P(Y, \hat{Y}) = \frac{TP}{TP + FP}$$

and $R$ stands for the measure of recall, given as

$$R(Y, \hat{Y}) = \frac{TP}{TP + FN}$$

Plugging this in Eq(1), and replacing the formulations for TP, FP and FN(from Eq A.2) we get

$$F\text{-}score(Y, \hat{Y}) = \frac{2TP}{2TP + FP + FN} = \frac{2(Y^T \times \hat{Y})}{\sum_{i=1}^n y_i + \sum_{i=1}^n \hat{y}_i} = \frac{2(Y^T \times \hat{Y})}{\mathbf{1}^T Y + \mathbf{1}^T \hat{Y}} \quad (11)$$

where $y_i$ refers to the $ith$ element of $Y$(same for $\hat{y}_i$). $\mathbf{1}$ represents an all one vector in $R^n$. Note that in training, since $Y$ is generally provided, we can assume $\mathbf{1}^T Y = \beta$ which is constant (the number of examples in the positive class in the ground truth). We can also represent the the values of $2Y$ as a coefficient matrix $c$, then the optimization problem for finding $F$-score can be written as

$$\begin{array}{ll} \underset{\hat{Y}}{\text{maximize}} & \dfrac{c^T \hat{Y}}{1^T \hat{Y} + b} \\ \text{subject to} & \hat{Y}_i \in [0,1], \ i = 1, \dots, n. \end{array} \quad (12)$$

### A.2.2 MAXIMIZING JACCARD COEFFICIENT AND $F_\beta$

The Jaccard Coefficient and Dice Index lead to similar formulation as $F$-score. The Jaccard coefficient can be expressed as:

$$\text{Jacc}(Y, \hat{Y}) = \frac{TP}{TP + FP + FN} = \frac{(Y^T \times \hat{Y})}{\sum_{i=1}^n y_i + \sum_{i=1}^n \hat{y}_i - \sum_{i=1}^n y_i \times \hat{y}_i} = \frac{(Y^T \times \hat{Y})}{\mathbf{1}^T Y + (1-Y)^T \hat{Y}} \quad (13)$$

This can be equivalently written as a linear factional program as shown in Model(10) where $c = Y$, $d = (1 - Y)$ and $b = \mathbf{1}^T Y$. The rest of the construction is similar to $F$-score.
Note that $F_\beta$ which is defined as

$$F_\beta(Y, \hat{Y}) = (1 + \beta^2) \frac{P(Y, \hat{Y}) \times R(Y, \hat{Y})}{\beta^2 P(Y, \hat{Y}) + R(Y, \hat{Y})} \quad (14)$$

where $\beta$ is a user specified parameter (balancing the importance of precision and recall) also permits a similar formulation. Here we simply set $c = (1 + \beta^2)Y$, $d = \mathbf{1}$ and $b = \beta^2 \mathbf{1}^T Y$.

### A.2.3 MAXIMIZING $P@R$

We begin by defining the maximum precision at fixed minimum recall problem as

$$P@R\alpha = \text{maximize} \quad P \quad \text{s.t.} R \geq \alpha \quad (15)$$

$$= \text{maximize} \quad \frac{(Y^T \hat{Y})}{\mathbf{1}^T \hat{Y}} \quad \text{s.t.} \quad Y^T \hat{Y} \geq \alpha \mathbf{1}^T Y \quad (16)$$

This is again a linear fractional objective with a linear constraint. So we can write it as an equivalent Linear program using the same transformation where $c = Y$, $d = \mathbf{1}$ and $b = 0$. $R@P$ on the other hand directly leads to a linear program.

### A.3 OPTIMIZING NON-NEGATIVE MATRIX FACTORIZATION (NMF)

Nonnegative matrix factorization is different from the other objectives we presented in that it is primarily used in unsupervised learning and does not satify the criteria for a metric loss function. It can still be formulated in a generalized non-decomposable form because **(i)** cannot be written as a sum over individual samples, **(ii)** leads to a model where the constraints depend on learned features.

Note that purpose of discussing NMF in this context, is not to provide a general purpose solver for the problem, and instead to assess whether NMF layers can serve as a regularizer or a clustering module, e.g., learning more interpretable attributes, co-segmentation and a substitute for clustering, see Trigeorgis et al. (2014) and Collins et al. (2018). The following description in this section and the experimental validation in the following section is a proof of principle instantiation of this idea.

We briefly review from Arora et al. (2012) and Recht et al. (2012), how NMF is written as a LP.

We know from Arora et al. (2012) that a NMF decomposition $V = FW$ where $F$ is $s \times s'$ and $W$ is $s' \times w$ and $V$ and $W$ have a row sum of 1, is 'separable' if the rows of $W$ are simplicial and there is a permutation matrix $R \in \mathbb{R}^{s \times s}$ such that $RF = \begin{bmatrix} I_r & M \end{bmatrix}^T$. The top $r$ rows of $F$ contains so-called anchor words. Recht et al. (2012) proposed a data-driven model where the most salient features in the data are used to express the remaining features, given as $V \sim CV$, where $C$ is of size $s \times s$. Assuming $V$ admits a rank-$r$ separable factorization, then $V = R^T \begin{bmatrix} I_r & 0 \\ M & 0 \end{bmatrix} RV = CV$. To show that thsis factorization is possible, we need to first make $F$ square, which is why it is zero-padded to make it a size $s \times s$ matrix. Let $\widetilde{p}$ be any vector which is used as the coefficient in the objective in the following model. According to Recht et al. (2012), any value for the entries of $\widetilde{p}$ should suffice as long as they are distinct. with distinct values. Then the LP formulation is as follows:

$$\min_C \widetilde{p}^T \text{diag}(C) \quad \text{s.t.} \qquad CV = V, \ \text{tr}(C) = r, \ C_{jj} \le 1 \ \forall j, \ C_{ij} \le C_{jj} \ \forall ij, \ C \ge 0 \qquad (17)$$

With $C$ in hand, $W$ is constructed by extracting rows of $V$ for those indices $k$ where $C_{kk} = 1$. $F$ is constructed by extracting rows of $C$ which correspond to $k$ where $C_{kk} = 1$.

### A.3.1 EXPERIMENTAL RESULTS ON NONNEGATIVE MATRIX FACTORIZATION

We demonstrate applicability of our strategy to nonnegative matrix factorization (NMF) by performing a rank $k$ factorization on Convolutional Neural Network (CNN) activations as an example, following Collins et al. (2018). Recall that the activation tensor of an image at some layer in CNN has the shape $V \in R^{c \times h \times w}$ where $h, w$ are the spatial sizes and $c$ is the number of channels. We can reshape it into $V \in \mathbb{R}^{c \times (h \cdot w)}$ and calculate a rank $k$ NMF for $V$: $V = FW$. Each row $W_j$ of the resultant $W \in \mathbb{R}^{k \times (h \cdot w)}$ can be reshaped into a heat map of dimension $h \times w$ which highlights regions in the image that correspond to the factor $W_j$. We show an example for $k = 1, 2$ in Fig. 2. We can see that heatmap consistently captures a meaningful part/concept in the examples. Currently, our memory consumption increases quickly with $c$ here since the constraint matrix in our LP formulation is of size $O(c^2) \times O(c^2)$. This makes our method only work for small $c$ on a GPU with 11GB memory (here, we use $c = 20$). This scaling issue can be possibly solved by utilizing sparsity in the constraint matrix, but the sparse matrix operations are currently not well supported on mainstream deep learning platforms like PyTorch and Tensorflow. Since our method provides backward gradients for the NMF operation, the heatmap generated here can, in fact, be used to construct a loss function during training in order to learn a interpretable models.

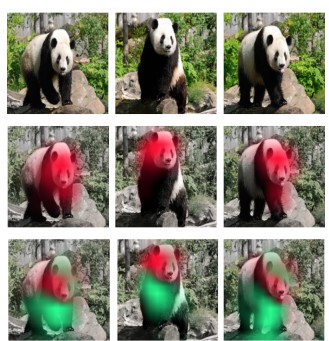

Figure 2: NMF example. Three rows correspond to original images, $k = 1$ and $k = 2$ respectively.

| Objective | $g$ | $h$ | $E$ | $F$ | $p$ | $B$ | $G$ | $q$ |
|---|---|---|---|---|---|---|---|---|
| AUC | $\mathbf{1} \in \mathbb{Z}^{\|T\| \times \|N\|}$ | - | $\mathbf{-1} \in \mathbb{Z}^{\|T\| \times \|N\|}$ | - | $p_{ij} = (f(x_i) - f(x_j) - \epsilon)$ | - | - | - |
| $\text{AUC}_\mu^{bin}$ | $\mathbf{1}^\dagger,$ | - | $\mathbf{-1}^\dagger$ | - | $p_{ij} = (\mathbf{f}_{ij}^\ddagger - \epsilon)$ | - | - | - |
| $F$-score | $\mathbf{c}$ | $0$ | $\begin{bmatrix} 1 \\ -1 \\ 1 \end{bmatrix}^\S$ | $\begin{bmatrix} -1 \\ \phi(f(x_i)) \\ -(1 + \phi(f(x_i))) \end{bmatrix}^\S$ | $0$ | $\mathbf{d}$ | $b$ | $b$ |

Table 5: Table showing the general LP coefficients for each model. $^\dagger$: length based on problem setting; $^\ddagger$: $\mathbf{f}_{ij}^\ddagger = \widetilde{\mathbf{d}}_{ij}(f(x_i, y_{C(x_i)}) - f(x_j, y_{C(x_i)}) + f(x_j, y_{C(x_j)}) - f(x_i, y_{C(x_j)}))$; $^\S$: one block for each $i \in [1, ..n]$. We do not include NMF in this table, as its formulation as a general LP is more verbose including vectorization of matrices and kronecker product calculations.

### A.4 VERIFICATION OF UNROLLING GRADIENT AND THE ONE PROVIDED BY $\tilde{A}^{-1}\tilde{b}$

We use Fscore formulation as an example. For input sample $x$, the neural network predicts a score $f(x)$, and then the scores of a batch of samples will be used in solving the linear programming form of Fscore and be used to construct the loss function. We compute the gradient from the final loss function back to the predicted scores from the neural network and compare two approaches: one is that we use $z = \tilde{A}^{-1}\tilde{b}$ as the solution (the one we used in our experiment) where we can compute gradient by only one step, another one is that we directly use $y_t$ resulting from the Newton iterations as the solution and compute gradients by unrolling those iterations. We then compute the cosine value between these two gradient vectors. By experiments on 100 randomly sampled batches, the average cosine value is $0.9991$, which means the two gradients are highly consistent.

## B PROOFS AND DETAILS OF RESULTS IN SECTION 3

In this section, we will provide the missing proofs and additional calculations in Section 3.

### B.1 PROOF OF LEMMA 1.

Lemma 1 is restated here for convenience.

**Lemma 3.** *Assume that $A \in \mathbb{R}^{m \times n}$ is a random matrix, and fix some $y \in \mathbb{R}^n$. Then with probability one, $g$ in equation 6 is quadratic (given by $\tilde{H}, \nabla g(y)$) over a sufficiently small neighborhood of $y$.*

*Proof.* Using the integral form of second order Taylor's expansion of $\sigma^2(y) = (\max(0, y))^2$, we can show that,

$$g(y+h) - g(y) - h^T \nabla g(y) = \frac{1}{2} h^T A^T \text{diag} (\mathfrak{d}) A h \tag{18}$$

where

$$\mathfrak{d} = \int_0^1 \left( \int_0^1 (\sigma (Ay - b))_* ds \right) 2dt. \tag{19}$$

See Remark 1 in Golikov & Kaporin (2019) for details. Without loss of generality, we can assume $b = 0$ by simply translating the origin. Following the same remark, the diagonal matrix coincides with the step function based diagonal in $\tilde{H}$ under the following condition on $h$:

$$e_j^T Ah \cdot e_j^T (Ay) < 0 \implies |e_j^T Ah| \le |e_j^T (Ay)|. \tag{20}$$

Since $y$ is fixed, assuming that the entries of $A$ are chosen from a continuous distribution such that $e_j^T A$ is uniformly distributed over the sphere, then $(e_j^T Ah)^2$ follows a Beta $\left(\frac{1}{2}, \frac{n-1}{2}\right)$ when $h$ is drawn uniformly at random from the unit sphere, independent of $A$. This means that no matter what $y$ is, there exists sufficiently small $h$ such that the left hand side of equation 20 is false with probability one, and in that neighborhood $\text{diag}(\mathfrak{d}) = \tilde{H}$. $\square$

### B.2 PROOF OF THEOREM 2

**Theorem 4.** *Assume that the primal LP has a unique optimal solution, and that the level set $\{x : Ax \le b, c^T x \le \alpha\}$ is bounded for all $\alpha$ (for dual feasibility). Then short step (no line search) Newton's method converges globally at a linear rate with local quadratic convergence.*

*Proof.* First, since the objective function is piecewise quadratic since it is a sum of piecewise quadratic functions. In particular, it is self concordant since its third derivative zero everywhere. Now setting $\rho < \epsilon$, we see that an approximate solution of the problem with the modified Hessian is also an approximate solution to equation 6. Moreover, since the possible values of $\tilde{H}$ is finite, the local norm (also known as Newton's decrement) $\nabla g(y)^T \tilde{H}(y)^{-1} \nabla g(y)$ is finite. Hence, we can choose $\rho > 0$ so that there is a descent direction $d$, that is, there exist a step size $\lambda > 0$ such that $\lambda \nabla g(x)^T d < 0$. Finally, we use Theorem 4.1.12 in Nesterov (2013) to claim the desired result. $\square$

The assumptions in Theorem 4 are standard: 1. uniqueness can easily be satisfied by randomly perturbing the cost vector; 2. in most of our formulations, we explicitly have bound constraints on the decision variables, hence level sets are bounded.

### B.3 DIFFERENTIATING THE STEP FUNCTION IN REMARK 4

We will use a slightly modified "suffix" notation as in Brookes (2005) in our calculations. That is, for a matrix $A$, $\vec{A}$ is the same as $\text{vec}(A)$, vectorization of $A$ obtained by concatenating all the columns. The following three properties relating the Kronecker product, $\vec{\cdot}$, and differentials will be used often:

1. Fact 1: For two vectors $a, b$, $a \otimes b = \overrightarrow{ba^T}$.

2. Fact 2: If $A$ is $p \times q$ matrix, and $B$ is a $m \times n$ matrix, then $\overrightarrow{\partial B} = (\partial B / \partial A) \overrightarrow{\partial A}$ where $\partial B / \partial A$ is the $(mn) \times (pq)$ Jacobian matrix of $\vec{B}$ with respect to $\vec{A}$. If $A$ or $B$ is a column vector or scalar, then $\vec{\cdot}$ has no effect.

3. Fact 3: $\overrightarrow{\partial (AXB)} = \left(B^T \otimes A\right) \overrightarrow{\partial X}$.

Using the above two facts, we can compute all the gradients needed to backpropagate through the unrolled iterations. We will show the computation for the gradient of $Q^{-1}u$ with respect to $A \in \mathbb{R}^{m \times n}$ for a fixed $u \in \mathbb{R}^n$. We can apply chain rule to the following composition:

$$
\begin{array}{ccc}
A & \xrightarrow{\ f_1 \circ f_2\ } & \left(A^T \tilde{H} A + \rho I\right)^{-1} u \\
f_1 \downarrow & \nearrow f_2 & \\
A^T \tilde{H} A + \rho I & &
\end{array}
$$

to get, $J_{f_2 \circ f_1} = J_{f_2} \circ J_{f_1}$. Now using Fact 2 on $\partial\left(X^{-1}\right) = -X^{-1}(\partial X)X^{-1}$ and some algebraic manipulation, we obtain,

$$
\overrightarrow{J_{f_2 \circ f_1}} = -\left(u^T \left(A^T \tilde{H} A + \rho I\right)^{-1} \otimes \left(A^T \tilde{H} A + \rho I\right)\right) \overrightarrow{J_{f_1}}. \tag{21}
$$

We will now compute $\overrightarrow{J_{f_1}}$. Note that $\tilde{H}$ is also a function of $A$, so using product rule, we can write $\overrightarrow{J_{f_1}}$ as a sum of three derivatives – with respect to each of $A, A^T, \tilde{H}$. The derivatives with respect to $A$ and $A^T$ are fairly straightforward to compute, so will focus on computing the derivative with respect to $\tilde{H}$. To that end, we will use Fact 3, and show to compute the derivative of the step function by approximating it using the logistic function.

$$
\frac{\partial}{\partial A}\text{diag}\left((Ay - b)_*\right) \approx \frac{\partial}{\partial A}\text{diag}\left(1 \oslash (1 + \exp\left(\kappa\left(-Ay + b\right)\right))\right), \kappa > 0. \tag{22}
$$

Note that these derivatives are used in computing derivatives of upstream network, so using distributional derivatives, and another application of chain rule to the left hand side of equation 22 results in the dirac delta function which is atomic, that is, has all its mass in a measure zero set. Hence this calculation provides an mathematical justification that the set of nondifferentiable points has measure zero for our training purposes. It is easy to formally verify this argument using differentiable tent functions as approximations to the heaviside step function.

