# OpenReview forum: "Differentiable Optimization of Generalized Nondecomposable Functions using Linear Programs"
_ICLR.cc/2021/Conference — Reject_

### Official Review · AnonReviewer2 · 2020-10-27
**Interesting ideas but not polished enough**

**Rating:** 3
**Confidence:** 5

**Review:**

Summary
-------

The paper makes the observation that various non-decomposable losses in machine learning can be rewritten as linear programs, whose constraints depends on the model output. This is the case for AUC, multi-class AUC, F-score, and to some extend NMF.

The authors review these losses, and recall how they may be rewritten as LPs. The LP formulation as known for AUC and NMF, but as far as the reviewer understand, they are new for multi-class AUC and F-score.

Then, the authors propose to directly backpropagate through the LP resolution to minimize non-decomposable losses, applied on top of deep architectures. For this, they propose to solve an approximate solution to the LP problem (a quadratic penalization of the constraint violations) using a modified Newton method. They propose either to backpropagate by unrolling the Newton steps, or by using the computed minimizer directly.

Review
------

The endeavor of writing non-decomposable losses as LPs, to see these losses as pluggable LP-layer in deep architecture is interesting, albeit not original.

Using a penalized approximation of the LP to be able to solve them efficiently using a Newton method is also interesting.

The experiment section shows that it is indeed beneficial to directly optimize over a certain decomposable loss when we measure performance in term of this loss: in particular, it outperform using a simple logistic loss. This was completely expected, but it is good to verify it experimentally.

On the other hand, the manuscript suffer from many unclear parts, and from a theoretical analysis that is not polished enough. In particular:

    - Phi is not introduced beforehand p. 4, and the F-score part is very hard to understand.

    - the NMF section is very unclear, in particular as the authors use vague terms in their construction, such as "zero padding ensures a sxs matrix". I do not understand the role of tilde p in (6).

    - Lemma 1 is not stated properly, as there is no f in equation (7). The authors state that "each y has a neighborhood in which the Hessian is quadratic", which does not mean anything. The proof sketch of Theorem 2 is very vague, in particular when the authors state that "the possible choice of Hessian is finite".

    - I do not understand whether rho is chosen at every iteration, and what is its importance.

I have trouble understanding why the authors went to such lengths in their
theoretical analysis. They modify a LP by making it a "smooth almost everywhere"
problem, which can then be solved using any methods, and backpropagated through
using either unrolling, or the computed minimizer (by virtue of Danskin
theorem), or the implicit function theorem. There is therefore not need to backpropagate throught tilde A^{-1} b.

The fact the the problem is only smooth almost everywhere may be a problem,
which is not addressed by using a Newton method. It implies that the gradient
becomes a subgradient, and may hinder optimization performance. Remark 4
dismisses this problem as unimportant, yet it is, as local convergence rates for
non-convex gradient descent requires smoothness.

Relating to experiments:

    - The reported performance does not show std errors across splits, which makes it impossible to compare in between similar methods (PPD-SG, PPD-AdaGrad and Ours). It appears that all three methods are within statistical variations.

    - NMF is a long studied problem, with many powerful methods to handle large inputs. I do not understand the choice of using the input of a deep learning network for the experiment. As it it, the experiment proposed in this manuscript is not polished enough to be valuable.

---

> ### Author Response · Authors · 2020-11-20
> **Response to R2's comments**
>
> We thank the reviewer for their comments and address the questions below.
>
> Q:  Phi is not introduced beforehand p. 4, and the F-score part is very hard to understand.
>
> Phi is introduced for the first time under Notations in Section 2.1. We apologize if the reviewer missed this. We have added additional clarification to math terms in this section.
>
> Q: NMF section is unclear: "zero padding ensures a sxs matrix". I do not understand the role of tilde p in (6).
>
> We deliberately tried to keep the NMF section brief since most of these details have been covered in detail in the Recht 2012 paper. We will remove this sub-section from the main paper and moves it to Appendix based on the reviewer’s suggestion.  A brief explanation is below,
>
> Assume that the matrix V=FW has two factors where $F$ is of size $s \times s’$. To show that a factorization of the form $V=R^T[I_r 0; M 0]RV$ is possible, we need to first make $F$ square, which is why it is zero-padded to make it a size $s \times s$ matrix. The vector $\tilde{p}$ is simply the coefficient in the objective. According to Recht 2012, any value for the entries of \tilde{p} should suffice as long as they are distinct.
>
> Q: Lemma 1 is not stated properly: no f in (7).
> The authors state that "each y has a neighborhood in which the Hessian is quadratic", which does not mean anything. Why "the possible choice of Hessian is finite".
> whether rho is chosen at every iteration
>
> We apologize for the typo in Lemma1. It should be g not f.
>
> We meant to say that "each y has a neighborhood in which the Hessian is constant". The proof is based on a classical result in Theorem 4.1.12 in Nesterov (2013) which established convergence rate for Newton's method globally.
>
> Possible choices of Hessian are finite because elements of A are finite.
>
> Rho is simply a hyperparameter that is fixed for all iterations.
>
> Q: Backpropagated through using either unrolling, or the computed minimizer (Danskin theorem), or the implicit function theorem. No need to backpropagate through tilde A^{-1} b.
>
> Notice that the Danskin’s theorem or implicit function theorem (IFT) depends on access to the optimal solution to the LP. Our approximation to the gradient is quite different from existing methods that use Danskin’s/IFT. To see this, note that our forward pass involves three steps: (i) finite steps of Newton’s method using which we  (ii) compute the dual variable by a thresholding operation, (iii) finally, to get the primal solution, these dual variables are first used to identify the active constraints followed by a linear system solve. We tried explaining this in the beginning of Section 3.2. Hence, in order to backpropagate through these three steps, we must differentiate through $\tilde{A}^{-1}b$ since that is the last layer of our procedure, independent of whether we use unrolling/Dankin/IFT for backpropagation purposes.
> One option to use IFT or Danskin in this framework is to simply to differentiate through the fixed point of the Newton’s iterations similar to (regularized) gradient descent iterations considered in the iMAML paper (https://arxiv.org/pdf/1909.04630.pdf). The details of the exact formula to do this is explained in text above Remark 4; we will make this clearer.
>
> Q:  The fact  the problem is only smooth almost everywhere may be a problem, which is not addressed by using a Newton method. It implies that the gradient becomes a subgradient, and may hinder optimization performance. Remark 4 dismisses this problem as unimportant.
>
> The function is a piece-wise quadratic function and differentiable everywhere. No adjustments to the gradient are needed. Rather, the inverse of the Hessian acts as a preconditioner.  Remark 4 does not dismiss this as unimportant and we have modified the text to avoid this confusion. In this perspective, Remark 4 simply provides a way to compute an approximate sub-gradient when using Newton’s method based LP layers.
>
> Q: The reported performance does not show std errors across splits.
>
> We are running these experiments now and will update it in the revision if time permits.
>
> Q:  Many powerful methods to handle large inputs in NMF. I do not understand the choice of using the input of a deep learning network for the experiment.
>
> We have removed this section but clarify below.
>
> First, several authors have explored using NMF within deep networks, for learning more interpretable attributes, co-segmentation and a substitute for clustering (Trigeorgis, ICML 14 and Collins, ECCV 2018). So, there is some value in investigating the role of NMF as a regularizer, a clustering module or to compare latent representations learned by a pair of generative models. The description was a proof of principle instantiation of this idea. Using a non-convex formulation of NMF here, solved in the standard way, would correspond to a noisy zeroth order oracle, which may be undesirable.

---

### Official Review · AnonReviewer1 · 2020-10-28
**Interesting ideas, experiments seem incomplete**

**Rating:** 6
**Confidence:** 2

**Review:**

This paper shows how some nondecomposable functions can be
interpreted as solving combinatorial optimization problems
that can be relaxed to linear programs.
As this paper brings some new insights and directions here
I recommend for a weak accept, although some of the experimental
settings and baselines feel incomplete (more details below).

# Strengths
Optimizing these performance metrics is useful for settings
where these are more important than optimizing for the accuracy,
and to the best of my knowledge these relaxations and ablations
meaningfully contribute to this direction.

The AUC experiments in Table 1 outperform [Liu 2019],
which also optimizes for the AUC.

The derivative computation can be computationally expensive
and ablating the impact of unrolling is interesting to
better-understand tradeoffs.

# Weaknesses
The biggest weakness I see is that the F-score experiment in
Table 3 has no baseline that also directly optimizes the F-score,
such as in [Fathony 2020] and other methods cited in the paper,
and the non-negative matrix factorization experiments are
lacking quantitative results.

This paper relies on relaxing the integer domain to be continuous
and it's not clear how much this approximation impacts the derivatives.

Some of the design choices seem arbitrary and unjustified, such as
focusing on the fast exterior penalty optimization at the start
of Section 3 and then doing the forward pass with Newton's algorithm
on the unconstrained problem from [Mangasarian 2004] in Section 3.1
to solve the LP.

# Other comments and questions
One recent related work is omitted: [Fathony 2020].

Is there any intuition behind why this approach sometimes outperforms
[Liu 2019] in Table 1, but sometimes doesn't? My interpretation without
this is that both of these approaches are approximations and it's hard
to know a-priori which will perform better.

Remark 2 says that in previous work, the LP constraints do not
depend on the data, but the formulations of [Amos 2017, Agrawal 2019]
allow constraints that depend on the data.

# References
Fathony, R. & Kolter, Z.. (2020). AP-Perf: Incorporating Generic
Performance Metrics in Differentiable Learning. Proceedings of the
Twenty Third International Conference on Artificial Intelligence
and Statistics, in PMLR 108:4130-4140

---

> ### Author Response · Authors · 2020-11-20
> **Response to R1's comments**
>
> We thank the reviewer for their comments and address the questions individually.
>
> Q: The biggest weakness I see is that the F-score experiment in Table 3 has no baseline that also directly optimizes the F-score, such as in [Fathony 2020] and other methods cited in the paper, and the non-negative matrix factorization experiments are lacking quantitative results.
>
> Based on the suggestion, we have added additional baselines to our experiments on Fscore. We thank the reviewer for pointing this out. We will add the reference in the main paper as well.
>
> Q: The non-negative matrix factorization experiments are lacking quantitative results.
>
> The NMF section has been removed based on reviewer suggestions. While this model was included to show a proof of principle example that our solver is applicable, showing compelling results in an interesting setting needs sparsity support for some operations (e.g., sparse linear system) that is not yet available in PyTorch.
>
> Q: This paper relies on relaxing the integer domain to be continuous and it's not clear how much this approximation impacts the derivatives.
>
> Note that the derivatives are being calculated with respect to the coefficients and the objective is a continuous function of the coefficients. Therefore, the rounding on $x$ does not impact the way derivatives are computed here. If the reviewer has a specific experiment that will be useful, we will add it.
>
> Q: Some of the design choices seem arbitrary and unjustified, such as focusing on the fast exterior penalty optimization at the start of Section 3 and then doing the forward pass with Newton's algorithm on the unconstrained problem from [Mangasarian 2004] in Section 3.1 to solve the LP.
>
> We do not see how these choices are mutually contradictory or arbitrary. Mangasarian’s method solves the parametric exterior penalty formulation of the primal LP, for a fixed value of the penalty parameter ε ≥ 0 precisely in the form of the unconstrained minimization problem, shown in (7) in Section 3.1. Therefore the two are intrinsically connected in this model and we will appreciate it if the reviewer can further clarify the question.
> In Section 3, we simply explain the advantages of this model over other existing methods, whereas Section 3.1 talks in detail about what optimization is solved in each step and its convergence properties.
>
> Q: Is there any intuition behind why this approach sometimes outperforms [Liu 2019] in Table 1
>
> Note that both ours and Liu 2019 maximize the AUC but the solution is obtained in different ways via different relaxations. One difference is that the authors in Liu 2019 use the squared loss as a surrogate for the indicator function in AUC, whereas our model can be thought of as closer to a truncated L1 loss.
>
> Q: Remark 2 says that in previous work, the LP constraints do not depend on the data, but the formulations of [Amos 2017, Agrawal 2019] allow constraints that depend on the data.
>
> We have modified this remark for clarity.
> Note that Amos 2017 (qpth) proposes a general purpose layer for convex optimization, not specifically designed for problems with a large number of constraints. Even though their model will work for our setting, it was primarily designed for QPs and has a cubic dependence of the run time on the number of variables and constraints. Agarwal’s method (Agrawal 2019), which is another effective approach for solving convex  optimization using a differentiable layer, is included in our experiments in this revision.

---

> > ### Comment · AnonReviewer1 · 2020-11-20
> > **Updates nicely address my initial comments**
> >
> > Thanks for the clarifications, edits, and new experiments! This nicely addresses all of my original comments. I am still feeling positive about this paper, but am still digesting the critiques and responses from the other reviewers.

---

### Official Review · AnonReviewer4 · 2020-10-28
**The method has merit but doesn't indicate it will succeed where so many others failed**

**Rating:** 5
**Confidence:** 5

**Review:**

This paper addresses the classical topic of directly optimizing non-decomposable loss functions. Since these metrics can be computed via linear programs, it is sufficient to compute gradient through the LP solver. To that end, the authors propose to use a particular method for solving linear programs. In experiments, the resulting implementation outperforms the cross-entropy loss and mildly outperforms one recent baseline.

The topic of direct optimization of AUC, F1, and all the other related metrics is still of high practical interest and meaningful progress would be of high significance. However, the main reason why this topic is still open is that none (or a handful) of the dozens of top-tier publications addressing it, delivered a method that is reliable, stable, and scalable to practical setups (e.g. in computer vision). For this reason, I think, any new paper in this area needs to undergo a high level of scrutiny.

I believe this paper has merit and constitutes a perfectly reasonable submission. At the same time, in its current form, it does not indicate sufficiently why it should have a different fate than all the other papers that also outperformed cross-entropy and subsequently disappeared.

My objections to the paper fall into three categories

### Conceptual confusion

In this section, I want to clarify that gradients to LPs are a mostly solved problem and there is no dire need to resort to a new type of an LP solver -- as the authors do. I will also contradict the claim "a simple closed-form gradient is not available for backpropagation" made on page 4

For clarity let's focus on three scenarios (in the notation of the paper):

1) Taking derivatives of the **optimal value of the objective function** w.r.t the LP parameters $c$, $b$, $A$

In all these cases, there is a simple closed-form gradient. E.g. if $x^*$ is optimal, the corresponding objective is $z = z(c) = c^T x^*(c)$ where the dependence of $x^*$ on $c$ is highlighted. This is a piecewise linear function ($x^*(c)$ is clearly piece-wise constant) with $dz/dc = x^*(c)$. This gradient $x^*(c)$ is simply the output of the LP and can be computed by any tailored method (not even necessarily by an LP e.g. by quicksort, Dijkstra, etc.) without requiring *any additional computation* on the backward pass.

The situation with $dz/db$ and $dz/dA$ is similar (via duality). This is all known for over 40 years ([M1, M2, M3, M4])

In the AUC LP formulation (eq 2), the resulting AUC score basically the negative optimal value of the objective, so **these classical results apply**.

2) Taking derivatives of the **optimal solution**  $x^*$ w.r.t the LP parameters $c$, $b$, $A$

In case of $dx^*/db$ and $dx^*/dA$ there are again simple closed-form gradients. One can either refer to eq (6) in (Amos, 2017)
 where closed-form gradients of more general QPs computed, or to an argument the authors make themselves in Sec 3.2. $x^*$ is a solution to the linear system given by the set of active constraints, and gradients of matrix inversion are easy to compute. These are also the gradients given by CVXPy -- even though admittedly their LP solver is currently very slow.

Most interesting is the case of $dx^*/dc$ as the true gradient is zero (dependence is piecewise constant) but this zero gradient is useless for optimization. Constructing meaningful gradient proxies is precisely the point addressed by (Berthet, 2020) but also earlier by [M5, M6, M7]. This is an ongoing research direction with competing methods.

3) The LP solver is the last layer in the neural net and the ground truth solution $x^*_{\textrm{true}}$ is available for supervision.

In this special case, which is actually the most common one in practice -- and also occurs in this paper, the situation is a lot easier. In fact, most works have focused on this scenario [M8, M9, (Song 2016)]. Also (Berthet, 2020) addresses it with the YF loss. The takeaway is that there are good loss functions for which a) the gradient can be computed only from the forward pass information b) robustness of the solutions can be enforced (e.g. by noise in (Berthet, 2020) or by a margin in M8).

In summary, there are suitable existing methods for efficient and often blackbox (not tied to a specific solver) gradient computation can be achieved. I find it imperative that the authors acknowledge this and compare to some of them.

### Inaccurate claims and omissions in related work

Claims about (Berthet, 2020)
- "approximate gradient" - The gradient is approximate for good reason (see above)
- "necessary to solve ~n perturbed LPs" - This is not true for the YF loss that can be applied at the end of the network (see above). There, in fact, only forward pass information is enough to obtain a gradient. Also, in the more general case (middle of the network) there are methods [M6] that need only one additional call to LP solver.
- "using off-the-shelf solvers ... can severely slow down" - a) SOTA LP solvers are extremely fast, b) if there is a faster "solver" for a concrete LP, such as quicksort for the LP formulation of ranking, (Berthet, 2020) allows using it. In summary, (Berthet, 2020 and M5) allow the usage of the fastest possible algorithm and by definition cannot slow down runtime.

Missing recent related work that also addresses nondecomposable metrics:
[M10, M11, M12, M13]

### Insufficient experimental evaluation

As far as I am concerned, the ideal experimental section for this paper would look like this:

a) more comparisons against recent literature on non-decomposable metrics (one recent baseline on one dataset is simply not enough)
b) Identifying that the source of the performance is the selected method for solving LPs -- by comparing to alternatives described above. In that case,  additionally producing qualitative explanation on why this is happening.
c) Find a fair way to report runtime and scaling with increasing instance sizes. Also comparing it to alternative approaches.

# Summary

My intention is not to dismiss the method, it seems to work reasonably well and has a formal backing, after all. But if this paper is to be of long-term value to the community, I have to insist on a major revision for the reasons described above.

### References

[M1] Gal, 1975, Rim Multiparametric Linear Programming

[M2] Freund, 1985, Postoptimal analysis of a linear program under simultaneous changes in matrix coefficients

[M3] De Wolf, 2000, Generalized derivatives of the optimal value of a linear program with respect to matrix coefficients

[M4] Gao, 2020, Differentiable Combinatorial Losses through Generalized Gradients of Linear Programs

[M5] Vlastelica, 2020, Differentiation of Blackbox Combinatorial Solvers

[M6] Ferber, 2020, MIPaaL: Mixed integer program as a layer

[M7] Wilder, 2019,  Melding the data-decisions pipeline: Decision-focused learning for combinatorial optimization

[M8] Tsochantaridis, 2005, Large Margin Methods for Structured and Interdependent Output Variables

[M9] Elmachtoub, 2017, Smart "predict then optimize"

[M10] Rolinek, 2020, Optimizing Rank-based Metrics with Blackbox Differentiation

[M11] Brown, 2020, Smooth-AP: Smoothing the Path Towards Large-Scale Image Retrieval

[M12] Fathony, 2020, AP-Perf: Incorporating Generic Performance Metrics in Differentiable Learning

[M13] Khim, 2020, Multiclass Classification via Class-Weighted Nearest Neighbors

---

> ### Author Response · Authors · 2020-11-20
> **Response to R4's comments-part1**
>
> Q) We thank the reviewer for their comments. Below we address the points raised by each reviewers individually.
> There is no dire need to resort to a new type of an LP solver -- as the authors do.
>
> As stated in the review, some of the off-the-shelf options with good algorithms and good implementations are currently slow. There is value in investigating if specific solvers will exhibit better performance traits and may mitigate scalability and other issues that the reviewer acknowledges remain outstanding.
>
> Q) I will also contradict the claim "a simple closed-form gradient is not available for backpropagation" made on page 4.
>
> Yes, we can see that the statement is incorrectly phrased. In this first paragraph of the section, we only wanted to point out that a computationally simple and scalable form of backpropagation is more challenging than other common layers/blocks used in deep learning, and is a topic that several papers have studied which we provide references to, right after this sentence.
>
> Q) Scenario 1-2: Taking derivatives of optimal value of the objective function w.r.t the LP parameters c, b, A. Taking derivatives of optimal solution x∗ w.r.t the LP parameters c, b, A. Classical closed form results or more recent papers apply.
>
> We do not dispute that other papers have shown how to backpropagate through a convex optimization module: our paper cites several such works including Amos et al, Meng et al and others. We will also include the other citations, including recent ones from 2020. Yes, classical results as well as results from more recent developments apply.
> Our main reason for choosing the Newton solver is because of its efficacy and computational efficiency when the number of constraints are large. While one can backpropagate through a general LP model, the ability to do so for a Newton solver has not been investigated and we believe is computationally beneficial for many problem settings in deep learning (where libraries have in-built support for GPU), some of which are included in our answers below.
>
> Q) In case of  dx*/db  and  dx*/dA there are again simple closed-form gradients. One can either refer to eq (6) in (Amos, 2017) where closed-form gradients of more general QPs computed, or to an argument the authors make themselves in Sec 3.2. Most interesting is the case of dx∗/dc as the true gradient is zero (dependence is piecewise constant) but this zero gradient is useless for optimization. Constructing meaningful gradient proxies is precisely the point addressed by (Berthet, 2020) but also earlier by [M5, M6, M7]. This is an ongoing research direction with competing methods.
>
> We agree to the facts specified by R4 here but not the conclusion. Indeed, both our method and Berthet constructs gradient proxies for the piecewise constant function.  The gradient proxy that our algorithm solves is an instantiation of Follow the Regularized Leader (FTRL) vs Follow the Perturbed Leader (FTPL) as used in Berthet et al -- see 1.4 and 1.8 in Perturbation Techniques in Online Learning and Optimization (https://ambujtewari.github.io/research/abernethy16perturbation.pdf )
> Clearly, both are equivalent from the perspective of online learning. But the reviewer will agree that the convergence behavior of our procedures are very different -- while the idea in Berthet et al depends on the measure of the optimal solution, it is the distance to the optimal solution for our procedure. While both these approaches are related theoretically, they can have very different properties in practice in the following sense. In LPs with a low volume feasible set, it is not at all clear whether perturbing the cost function provides a useful approximation (even when the variance used is arbitrarily small), that is, in this case, the distance between the optimal solutions of the perturbed and original problem is equal to the diameter of the feasible set. In practice, this case happens when the constraints are also generated on-the-fly and depend on data, which is exactly the type of problems we have described in the paper. To our knowledge, this is not discussed in Berthet et al. On a separate note, it is also possible to combine best of both the worlds: use randomization along with exterior penalty.
>
> Q) Takeaway is that there are good loss functions for which a) the gradient can be computed only from the forward pass information b) robustness of solutions can be enforced (e.g. by noise in (Berthet, 2020) or by a margin in M8).
>
> Indeed, our paper also utilizes the fact that only forward pass information is required to compute the derivative due to implicit function theorem. Robustness to noise is in an expectation sense, which our method of course inherits from the regularization perspective. If loss function is sensitive (non-linear or sharp) there may be error in the gradient which may be magnified.

---

> > ### Author Response · Authors · 2020-11-20
> > **Response to R4's comments-part2**
> >
> > Q)    Inaccurate claims and omissions in related work
> >
> > Q.1) Claims about (Berthet, 2020): "approximate gradient" - The gradient is approximate for good reason (see above)
> >
> > We hope the above explanation clarifies this. Both our approach and Berthet et al provide different approximations for the LP. While they treat LP solvers as a black box quite effectively, we believe that the implementations in the current state of the art solvers that can fully run on a GPU within popular deep learning libraries remains limited. This is a key property of our algorithm that enables use in scalable settings that the reviewer notes remains a pain point. We believe this is a key contribution. One important example where GPU operation is significantly faster than CPU operation is the mini-batch operation widely used in almost all deep learning applications. The GPU based framework (e.g. Pytorch, Tensorflow) usually support batch operation where the mini-batch size has almost no influence on the running time. However, for most CPU based solvers, batch operation is not supported which mean one need to call the solver $B$ times when the mini-batch size is B. At the same time, mini-batch size is usually chosen to be large especially in large-scale experiments (e.g., MSCOCO) and the benefit of batch operation is especially significant at such settings.
> >
> > Q.2) "necessary to solve ~n perturbed LPs" - This is not true for the YF loss that can be applied at the end of the network (see above). There, in fact, only forward pass information is enough to obtain a gradient.
> >
> > To our knowledge, the position of the LP block does not change the computational properties from the perspective of the Fenchel Young losses as defined in Berthet et al., under Definition 4.1 is computationally attractive because there is no need to compute the Jacobian. From the implementation standpoint, one could simply think of the backward pass as a function given the input and output of the forward pass.  R4 will see that the gradient expressions (in the same section) of the FY losses involves the y^*_{\epsilon}(\cdot) which is an expectation from Prop 3.1 and hence requires multiple calls in order to approximate the expectation.
> > The text before Section 4 in that paper notes that with parallelization and warm starts, one can alleviate the dependency in M by sampling in parallel, although M=1 can be sensible in some settings (see Fig 2 in that paper).  But this will likely depend on properties of the LP, as we explained above.
> >
> > Q.3) Also, in the more general case (middle of the network) there are methods [M6] that need only one additional call to LP solver.
> >
> > The Newton layers can also be used in the middle of the network. It appears that [M6] is not accompanied with code but we will try to include a comparison, if time permits.
> >
> > Q.4) a) SOTA LP solvers are extremely fast, b) if there is a faster "solver" for a concrete LP, such as quicksort for the LP formulation of ranking, (Berthet, 2020) allows using it. (Berthet, 2020 and M5) allow the usage of the fastest possible algorithm and by definition cannot slow down runtime.
> >
> > Yes, it is indeed possible to solve the LP using external solvers. However, mathematical optimization solvers like Cplex and Gurobi still lack support for GPUs, unlike our algorithm. So, while the rest of the training proceeds on the GPU, there is a memory overhead to transfer data between the GPU and CPU to solve the LP using an external solver. To our knowledge, this issue cannot be easily avoided unless the off-the-shelf solver also runs natively on the GPU.
> >
> > Besides as mentioned in the previous response, Berthet will require M calls (M5 requires one call) to such a solver, it is not clear how large $M$ needs to be for a general purpose LP with a large number of constraints. The papers do not have experimental results to that effect. The code from Berthet et al will be open-sourced in TF but is not publicly available yet making direct comparisons difficult.

---

> > > ### Author Response · Authors · 2020-11-25
> > > **Comparison with potential SOTA LP solvers**
> > >
> > > Dear R4,  Based on your suggestions, we performed comparisons between our solver and approaches that can easily utilize a powerful SOTA LP solver in the forward/backward pass (e.g., M5, M6). This solver will utilize CPU operations.
> > >
> > > We performed experiments to evaluate potential benefits and limitations. We compared the runtime between our solver and the solver from the scipy package for LPs arising from our AUC problem (single 2080TI GPU and 20 core i9 CPU). We construct an example with 40 positive samples and 40 negative samples, corresponding to a batch size B. When B=1, our solver requires 0.52 (s) and scipy needs 0.1(s). So, we are much slower. When B=30, our solver needs ~2.05 (s) and scipy needs ~3.01 (s). This is because GPU supports batch operations naturally. Considering that the time spent on LP solving starts to dominate over the time spent on the forward pass of Resnet, it means that our solver can save one third of the total training time which is often multiple hours. Further, this benefit keeps increasing when the batch size further increases (given a GPU with a larger memory or libraries with moderate support for sparse operations). When B=60, our solver saves one half of the training time compared with scipy. We also conducted experiments using the code from M5 which solves the traveling salesman problem by calling Gurobi, and we observe a similar trend there: the runtime is typically B times more when the batch size increases from 1 to B.

---

### Official Review · AnonReviewer3 · 2020-10-30
**A good solution to tackle down large linear programming for training, but with some biased experiment settings.**

**Rating:** 5
**Confidence:** 4

**Review:**

A. Summary:
This paper approximates several nondecomposable functions (AUC and F1-score) as linear programmings and uses them as loss functions for network training. In the linear programmings, the constraints are indeterministic at each mini-batch, the number of constraints increases quadratically to the number of training samples, so some previous works are inapplicable here. So does the primal-dual based forward pass and the corresponding implicit differentiation for a backward pass.
Instead, the authors propose to select Newton's method for the forward pass and an adaptive online adjustment for the backward pass, where gradients are back-propagated through unrolled optimization before a proper accuracy is found, and then through complementary slackness afterward.
Finally, the experiments emphasize the condition when the positive/negative examples are imbalanced, and demonstrates the superiority of the proposed methods to cross-entropy loss and one previous AUC loss.

B. Strength:
1. This paper gives a clear discussion about the difference between the proposed method and previous works, which motivates the author to explore an alternative solution to integrate linear programming as a differentiable component for training.
2. It is a good choice to use Newton's method to tackle down the problem for training. For training, we care less about whether an optimization algorithm will achieve the global optimal solution, instead, we care more about whether the algorithm can achieve a local optimal rapidly, therefore the training will be feasible for larger-scale problems while the gradient can still be derived through complementary slackness.

C. Weakness:
1. Determining the accuracy \epsilon seems to be critical for the proposed algorithm. It would be good if the author could have some more explanation about this. Besides, is that possible to make it a learnable variable through back-propagation at the first stage? Since the gradient is backpropagated via unrolling at first, it should be doable and save some manual effort to decide how to increase it.
2.  In the experiments, the authors constructed an imbalanced dataset from the existing dataset, In addition, some other recent works, such as Lin, TY, Goyal, P, Girshick, R, He, K, Dollar, P, "Focal Loss for Dense Object Detection", ICCV'2017, also focus on deal with the class imbalance by other tricks. So it would be better to compare with these methods as well but not only vanilla cross-entropy.
3.  The application of nonnegative matrix factorization should be further improved. As indicated in the paper, it only works when the number of channels is limited. The author may remove this application and focus on the AUC and F1-score. The contribution will be sufficient enough if these two widely used metrics can be discussed more solidly.
4.  What if we fine-tune a pre-trained model using the proposed losses and the dataset is not manually selected to make it imbalance? Will the performance achieve consistent improvements?

D. Justification of the score:
In general, this paper focus on an important problem, and the algorithm is discussed comprehensively. My major concern is about the experiments, where the experimenting settings are biased to the proposed algorithm. I will raise my score if these concerns can be addressed during rebuttal.

---

> ### Author Response · Authors · 2020-11-20
> **Response to R3's comments**
>
> We thank the reviewer for their comments. Below we address the points raised by each reviewers individually.
>
> Q:  Determining the accuracy $\epsilon$ seems to be critical for the proposed algorithm. It would be good if the author could have some more explanation about this.
>
> To choose $\epsilon$, we follow the approach proposed by Mangasarian 2004. If for two successive values of $\epsilon_1 > \epsilon_2$, the value of the $\epsilon$ perturbed quadratic function is the same, then it is the least 2-norm solution of the dual. Therefore, we simply choose an $\epsilon$ that satisfies this property. We have added one paragraph in our revised version (in experiment section) describing this, including an ablation study of different $\epsilon$. We are not sure at this time whether learning it is possible.
>
> Q:  In experiments, the authors constructed an imbalanced dataset from the existing dataset, Better to compare with Lin et al other than vanilla cross-entropy.
>
> We agree with the suggestion of the reviewer and are working on this experiment and if time permits, will include it in a revision shortly.
>
> Q:  The application of nonnegative matrix factorization should be further improved. As indicated in the paper, it only works when the number of channels is limited. The author may remove this application and focus on the AUC and F1-score. The contribution will be sufficient enough if these two widely used metrics can be discussed more solidly.
>
> We appreciate the suggestion and have removed this sub-section; yes, this is not a main focus of the paper and we defer this short discussion on applicability to NMF in the Appendix.
>
> Q:  What if we fine-tune a pre-trained model using the proposed losses and the dataset is not manually selected to make it imbalance? Will the performance achieve consistent improvements?
>
> We are running experiments to address this comment, and will update the revision once the runs have completed.

---

> > ### Author Response · Authors · 2020-11-25
> > **About imbalanced dataset**
> >
> > R3 suggested that we discuss the performance of our solver on a naturally imbalanced dataset. We conducted experiments on celebA which includes a set of binary attributes and some of these are fairly imbalanced. We used “Receding Hairline” as an example where the positive:negative ratio is 1:11. After optimizing AUC directly using our solver, the AUC metric is 92.88% and behaves similarly as optimizing the cross-entropy loss (92.75%). Even though the imbalance may appear high, the minority class (people with hairline receded) includes sufficient number of samples for the model to learn.  This suggests that the benefits of optimizing the AUC will depend on the application. Indeed, by definition, in cases where it is easy to control the False Positivity Rate of the predictor with respect to the minority group, cross entropy and AUC behave similarly and so optimizing one will benefit the other implicitly. Of course, AUC may not be a drop-in replacement for any class imbalance correction procedure such as focal loss since these methods do not have any connections to FPR type quantities. Our experiment suggests that our formulation can reliably optimize AUC in these vision settings (which R4 commented would be desirable).

---

### Decision · Program_Chairs · 2021-01-07
**Final Decision**

**Decision:**

Reject

**Comment:**

This paper shows that various discrete loss functions can be formulated as an LP. It proposes to relax the constraint Ax = b, x >= 0 using a soft constraint and following Mangasarian, proposes to solve the relaxed problem using Newton's method. Backpropagation through these iterations is further proposed. The main motivation is that this results in a GPU-friendly implementation.

I think the proposed approach is novel. However, as pointed out by reviewers, the current writing lacks clarity and the experiments are quite weak. There is now a wealth of methods for differentiating through an LP using implicit differentiation, smoothing (which the present paper is a form of, see below) and perturbations. It is important to compare to these methods. The paper also ignores a large literature on convex surrogates for ranking metrics.

I recommend the authors to strengthen the writing and experiments, and to resubmit to a top-conference.

Additional comments by the AC
-------------------------------------------

As the sentences "Hence, solving such LPs using off-the-shelf solvers may slow down the training process" or "Often, this would involve running the solver on the CPU, which introduces overhead" indicate, the authors seem to imply that LPs need to be solved in canonical LP form, min_x <c,x> s.t. Ax <= b, x >=0, using an off-the-shelf LP solver. This is not how many LPs are solved in practice. For every loss, there will always be an ad-hoc solver for the corresponding LP. For instance, the Hungarian algorithm for the Birkhoff polytope.

The paper is missing an important reference: SparseMAP (https://arxiv.org/abs/1802.04223). In this paper, the authors add regularization to the primal LP and use Frank-Wolfe or active set methods to solve the problem.

Equation (6) corresponds to relaxing the hard constraint Ax=b, x>=0 with a soft one. This approach is in a sense opposite to SparseMAP. Indeed, relaxing the constraints in the primal is equivalent to adding regularization in the dual LP (see, e.g., https://papers.nips.cc/paper/2012/hash/bad5f33780c42f2588878a9d07405083-Abstract.html).
Speaking of (6), the authors should clarify that it's a convex objective.

In section 2, the authors review a number of losses which can be written as an LP. It would be better to explicitly state what are A, b and c for each loss (or g, h, E, F, p, B, G, q).

The matrix A could potentially be huge, depending on the LP. Do you need to materialize it in memory in practice? This would limit the approach to relatively small LPs.